
# Daily INSOLation (DINSOL-v1.0) model: An intuitive tool to be coupled with climate models and used in classrooms

Emerson Damasceno Oliveira

Laboratory of Meteorology, Federal University of Vale do São Francisco (UNIVASF), 48902-300, Juazeiro-BA, Brazil

**Correspondence:** Emerson D. Oliveira (emerson.oliveira@univasf.edu.br)

**Abstract.** Climate modelling requires spending an extensive amount of time programming, which means reading, learning, testing, and evaluating source code. Fortunately, many climate models have been developed within the past decades, making it easier for climate studies to be conducted on a global scale. However, some climate models have millions of code lines, making the introduction of new parameterizations a laborious task that demands teamwork. While it is true that the high com-

plexity models perform realistic climate simulations, some researchers perform their studies using simplified climate models in the preliminary test phases. This realisation motivated the development of the Daily INSOLation (DINSOL) model, a robust computer program to support the simplified climate models, performing solar radiation calculations while considering Milankovitch cycles and offering various simulation options for its users. DINSOL was intended to function as a model which supplies data (e.g., daily insolation, instantaneous solar radiation, orbital parameters of the Earth, and the calendar dates), such

as the PMIPII. While preparing the boundary conditions of solar radiation for climate models, it was realised that the DINSOL model could also be a helpful tool for use in classrooms. Thus, it was decided that an intuitive graphical user interface would be required to cater to this educational purpose. The model was written in the Fortran 90 language, while its graphical user interface would be built using PyGTK, a Python application programming interface (API) based on GIMP ToolKit (GTK). Furthermore, the R language would also be used to generate a panel containing contour fields and sketches of the orbital pa-

rameters to support the graphical execution. The model evaluation made use of data from PMIPII and other models, and the data analysis was performed through statistical methods. Once all tests were concluded, an insignificant difference between the DINSOL-obtained results and the results obtained from other models validated the viability of DINSOL as a tool.

## 1 Introduction

In paleoclimatology, greenhouse gases (GHGs) and other climate features, such as air temperature, can be estimated using

indirect methods (e.g., ice cores, speleothems, tree rings, lake and marine sediments, glacier fluctuations, and others) (Klippel et al., 2020). Paleoclimatology also investigates the effect that changes in the orbit of the Earth has on the incoming solar radiation (ISR). GHGs are vital because they affect the net radiation by increasing or decreasing the heat trapped in the atmosphere. Therefore, fluctuations in ISR or GHG affect the global energy balance, meaning that they are factors in global climate change (Menviel et al., 2019; Lhardy et al., 2021). Berger (2021) details how a century ago, Milutin Milankovitch

proposed a revolutionary approach to explaining the quasi-periodic occurrence of ice-ages from caloric season measurements.





It was because of his contributions that he has been considered as the father of paleoclimate modelling. Thus, the conceptual climate model developed and adopted by Milankovitch assumed that ISR changes happened due to cyclic oscillations of the Earth's Orbital Parameters (EOP): obliquity, eccentricity, and precession of the equinox. From Puetz et al. (2016), the initial Milankovitch theory was treated with scepticism due to previous theories on what causes ice-ages, theories relating
to the ejection of volcanic dust content in the atmosphere, as well as the cyclic changes in the magnetic field of the Earth.

Berger (2021) states that the most important books published by Milankovitch are *Théorie mathématique* (1920) and *Kanon der Erdbestrahlung* (1941). In these books and other literature, Milankovitch used EOP data previously calculated, adopting Stockwell–Pilgrim values from the theoretical investigations on ISR and ice-ages. From Crucifix et al. (2009), besides Milankovitch; other authors investigated the relationship between ice-ages and ISR, the most distinguished being André Berger,
who developed a practical method to calculate the EOP from trigonometric series. He also adopts the caloric seasons to investigate the past climate as a function of the EOP. In the 1970s, Berger published papers (e.g., Berger, 1976, 1977, 1978a, b) that consolidated the modern concepts of ISR modelling. Since then, novel solutions for Earth's orbit cycles have been developed (e.g., Laskar, 1988; Berger and Loutre, 1991; Laskar et al., 1993, 2004, 2011).

Presently, the Paleoclimate Modelling Intercomparison Project (PMIP) represents the best efforts of the scientific commu-
nity in paleoclimate reconstructions. PMIP is in its fourth phase (PMIP4), with some studies ongoing and others already been published (e.g., Jungclaus et al., 2017; Otto-Bliesner et al., 2017; Kageyama et al., 2018; Menviel et al., 2019). From Messori et al. (2019), most climate model simulations showed the intensification and geographical expansion of the monsoonal precipitation during the mid-Holocene, 6 Kyr before present, as suggested by proxy evidence. Therefore, climate reconstructions are crucial to enhancing the hold on understanding of natural forcings. Moreover, the Coupled Model Intercomparison
Project (CMIP) is capable of handling various scenarios, considering natural and anthropogenic forcings, and reducing the uncertainties of climate projections (Eyring et al., 2016). Further, the Intergovernmental Panel on Climate Change (IPCC) is responsible for announcing the CMIP scenarios through assessment reports, focusing mainly on providing readable summaries for policymakers (Fischer et al., 2020).

In recent years, useful tools to aid in calculating the EOP and insolation were developed, tools like the PALINSOL, an R
package written by Michel Crucifix that adopts Berger (1978b), Berger and Loutre (1991), and Laskar et al. (2004) solutions. Another tool is the Earth-Orbit v2.1, a MATLAB program created by Kostadinov and Gilb (2014) to calculate the EOP according to the Berger (1978b) and Laskar et al. (2004) methodology. The PALINSOL and other similar Fortran programs written by André Berger are available to download by the Université Catholique de Louvain (UCLouvain) through the Earth and Life Institute web page (https://www.elic.ucl.ac.be/modx/index.php?id=83). Even though we have pre-existing programs to calcu-
late the ISR following the Milankovitch cycles theory, it is remarkable that none was developed to prepare ISR data flexibly, such as a model. In this way, it would be interesting to have some software that performs all tasks as any pre-existent tool but additionally allows us to prepare custom solar radiation data, in other words, facilitates the process of generating insolation boundary conditions.

To achieve this goal, the Daily INSOLation (DINSOL-v1.0) model was developed, intended to be an intuitive and versatile
tool ideal for paleoclimate simulations, such as those performed on the PMIP. Additionally, DINSOL can calculate the ISR for





hypothetical cases, making it a viable option for exoplanet climate modelling. Users have various input choices to run their DINSOL model simulations (e.g., year, solar constant, latitudinal and longitudinal number points). It also includes a graphical user interface (GUI), with the GUI written in PyGTK, a Python program that adopts the GTK library. In addition, the DINSOL source code is mostly a Fortran program because all its important processes (e.g., data reading, take decisions, calculation, and
result writing) were written using Fortran 90 language. Besides PyGTK and Fortran 90, the model also contains some script templates written in R language to assist its users in accessing simulation results. It is worth pointing out that the program has a GNU GPL-v3.0 license, which allows users to modify, share, and improve it. For instance, the DINSOL was adapted to prepare ISR data for an energy balance model used by Oliveira and Fernandez (2020). Finally, the next chapters describe the model source code, output a detailed explanation of the astronomical equations, ISR and EOP parameterizations, input, and
output data, as well as a model evaluation chapter, where we adopted the PMIPII data as a reference.

## 2   Astronomical aspects and model description

This section details the astronomical fundamentals required to understand the DINSOL source code. The variables, constants, and mathematical relations are explained in the subsequent subchapters.

### 2.1   Main orbital elements and some formulas

The DINSOL model utilises heliocentric coordinates, such as in Berger (1978b) and Berger et al. (2010), and the most important information on the Earth's orbit shown in the Figure 1. Below, we have some orbital elements and constants for current days:

$$S_0 = 1366 \ W/m^2$$
$$T = 365.2422 \ days/year$$
$$a \backsim 150 x 10^6 \ km$$
$$e = 0.016724$$
$$\varepsilon = 23.446°$$
$$\varpi = 282.04°$$
$$\omega = 102.04°$$

where the solar constant, $S_0$, the tropical year, $T$, the semi-major axis, $a$, and eccentricity of the Earth's elliptical orbit, $e$. The longitude of the perihelion is kept unchanged, $\varpi$, given from vernal point (March 21) until perihelion day, and the longitude of the perigee, $\omega$, that is $\varpi$ added to 180°. As shown in Figure 1, the Earth's orbital revolution occurs counterclockwise, while the equinoxes precession occurs clockwise.



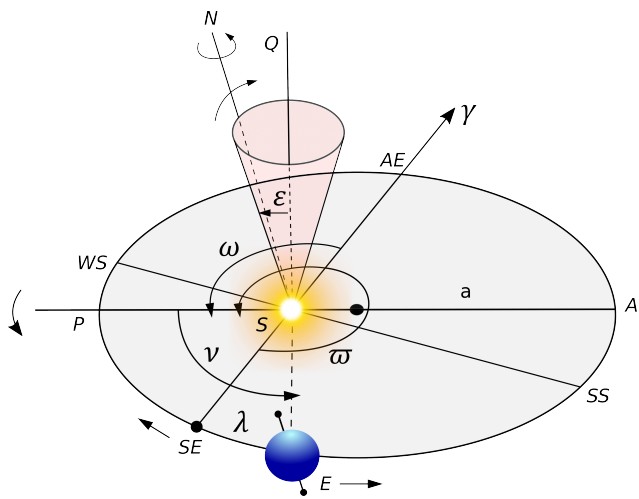

**Figure 1.** Heliocentrical coordinates sketch based in Berger (1978b) and Berger et al. (2010). The orbital elements are denominated of Earth, $E$, Sun, $S$, the semi-major axis of the Earth's orbit, $a$, perihelion, $P$, aphelion, $A$, vernal point, $\gamma$, spring equinox, $SE$, summer solstice, $SS$, autumn equinox, $AE$, winter solstice, $WS$, longitude of the perihelion, $\varpi$, longitude of the perigee, $\omega$, true solar longitude, $\lambda$, true anomaly, $\nu$, obliquity, $\varepsilon$. The SQ is perpendicular to the ecliptic, and $N$ is the north pole.

Regarding Figure 1, the Earth's obliquity representing the equatorial plane, $\varepsilon$, given by a perpendicular cone to the ecliptic plane, the Earth-Sun distance, $r$, measured in units of the semi-major axis, $a$, been given by the ellipse equation:

$$\rho = \frac{r}{a} = \frac{1 - e^2}{1 + e \cos \nu} \tag{1}$$

where the relative Earth-Sun distance, $\rho$, provides values for an annual calendar, such as the Figure 2. The true anomaly, $\nu$, is measured counterclockwise from the perihelion and given by the equation $\nu = \lambda - \varpi$, where $\lambda$ represents the true solar longitude. Equation 1 is given in Beutler (2005, p. 127).

The DINSOL model implements the methodology from Berger (1978b), which solves $\lambda$ in a few steps assuming that the start day is the vernal equinox (March 21), where $\lambda = 0$. Thus, we first need to finding the mean longitude $\lambda_m$, however, before is necessary to calculate the $\lambda_{m0}$ using the equation 2, where $\beta = \sqrt[2]{1 + e^2}$ and $e$ is the eccentricity.

$$
\begin{aligned}
\lambda_{m0} = \lambda - 2 \Big[ & \left( \frac{1}{2}e + \frac{1}{8}e^3 \right) (1 + \beta) \sin(\lambda - \varpi) \\
& - \frac{1}{4}e^2 \left( \frac{1}{2} + \beta \right) \sin 2(\lambda - \varpi) \\
& + \frac{1}{8}e^3 \left( \frac{1}{3} + \beta \right) \sin 3(\lambda - \varpi) \Big]
\end{aligned}
\tag{2}
$$

Finally, $\lambda$ is calculated using equation 3, implementing a loop that solves $\lambda$ along one year. However, for the first day (first iteration), the model requires $\lambda_{m0}$ value obtained previously from equation 2 where we assume $\lambda_m = \lambda_{m0}$, to be employed. Hence, in the proceeding days, $\lambda_m$ must be calculated from a simple increment equation given by the formula



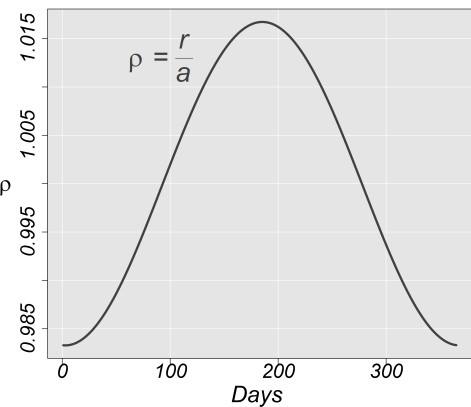

**Figure 2.** Relative Earth-Sun distance, $\rho$, along one year in Astronomical Unit (AU) for current days.

$\lambda_{m(i)} = \lambda_{m(i-1)} + 360/N_d$, where $N_d$ represents the number of days within one year. In the DINSOL, two annual calendars are available, a 365-day and 360-day calendar.

$$\lambda = \lambda_m + \left(2e - \frac{1}{4}e^3\right)\sin\left(\lambda_m - \varpi\right) + \frac{5}{4}e^2\sin 2(\lambda_m - \varpi)$$
$$+ \frac{13}{12}e^3\sin 3(\lambda_m - \varpi) \tag{3}$$

Figure 2 represents the annual variation of Earth-Sun distance, $\rho$, for the present day. Furthermore, the data used to plot $\rho$ was simulated using DINSOL, solving equations 1 to 3. A calendar conversion was initialised with a starting date of January 1st instead of March 21. The subsequent chapter contains all the details of the calendar conversions in the source code.

## 2.2 Defining the calendar dates

The daily insolation algorithm used in Berger (1978b) uses classical astronomical equations where the year is initialised from spring equinoxes, like the solar Hijri calendar, also known as the Persian or Iranian calendar. The Persian calendar accurately calculates the length of a season due to the use of the true solar longitude, $\lambda$, which works on elliptical coordinates, like in Keppler's laws. Although Berger (1978b) assumed March 21 as the fixed date for spring equinoxes, season dates oscillate over the years. For instance, in Borkowski (1996), in the second half of the 21st century, the spring equinoxes occurred between March 19 and 20. An online program offered by NASA can accurately calculate the date of astronomical events using the Gregorian calendar (https://data.giss.nasa.gov/modelE/ar5plots/srvernal.html, last access: 17 July 2022). The Gregorian calendar is lunisolar and assumes the temporal definition of months and seasons, where the astronomical dates change slowly (Joussaume and Braconnot, 1997). For instance, considering $\lambda$, the assumption is that seasons stay at 90° from one another along an elliptical orbit. Furthermore, a common challenge for paleoclimate simulations is the comparison of past and present climates, considering the differences between calendars.



From Bartlein and Shafer (2019), the number of days in a month are not constant, which means that the first day of each month might occur in a different position relative to the current Gregorian calendar. For instance, in (Joussaume and Braconnot, 1997), during the Eemian periods, the January month should have 34 days and start on December 25 compared to our current calendar. PMIPII assumes the vernal equinox as the time reference, keeping the current format of the Gregorian calendar for any period, which allows variations in seasons length, as well as changes in the aphelion and perihelion dates. From this method, the models can compare the results of paleoclimate simulations by using the same calendar structure, assuming a standard format. PMIPII experiments use a 365-day and 360-day calendar year, while typical climate models use a 360-day calendar.

The astronomical event dates (seasons, perigee, and apogee) are functions of the changes in two orbital parameters: eccentricity and precession of equinoxes. Furthermore, it must be noted that DINSOL uses a 365-day and 360-day calendar. Consequently, it must be considered that the vernal equinox (March 21) always occurs on the 80th day of a 365-day calendar and on the 81st day of a 360-day calendar. Finally, a modified version of equation 2 must be implemented assuming a simple calendar conversion (solar to Gregorian calendar), where estimation of the perihelion and aphelion dates will be made using the equation below:

$$
\begin{aligned}
P_d = -\varpi - 2 \Bigg[ & \left( \frac{1}{2}e + \frac{1}{8}e^3 \right)(1+\beta)\sin(-\varpi) \\
& - \frac{1}{4}e^2 \left( \frac{1}{2}+\beta \right)\sin 2(-\varpi) \\
& + \frac{1}{8}e^3 \left( \frac{1}{3}+\beta \right)\sin 3(-\varpi) \Bigg]
\end{aligned}
\tag{4}
$$

where $P_d$ represents the perihelion day on a solar calendar, and $P_d$ can be converted for a day in the Gregorian calendar if equation 5 is used. The assumption that the vernal point (March 21) is called $M_{21}$, which accepts the values 80 or 81 in concordance with the chosen calendar, $N_d = 365$ or $N_d = 360$, respectively.

$$
P_d = M_{21} + |P_d| \frac{N_d}{360}
\tag{5}
$$

After obtaining $P_d$ for a day of the year in the Gregorian calendar, the aphelion day can be determined, $A_d$, adopting the equation below:

$$
A_d = P_d + \frac{N_d}{2}
\tag{6}
$$

The DINSOL model has a subroutine that converts the day of the year to its correspondent month and day. Moreover, beyond perihelion and aphelion dates, the start date of a season can be determined. A fixed date for the spring equinox (March 21) must be assumed, $\lambda = 0°$. Then, using the true solar longitude, $\lambda$, the Summer, Autumn, and Winter start dates are known, at $\lambda = 90°$, $\lambda = 180°$, and $\lambda = 270°$ respectively. Then, conform was discussed in section 2.1, $\lambda$ must be solved using a loop to determine the iterations corresponding to $\lambda$ equal to 90°, 180°, and 270°. Furthermore, the DINSOL model uses a two-decimal precision, like in the PMIP II project. The calendar function evaluation is provided in subchapter 3.2.



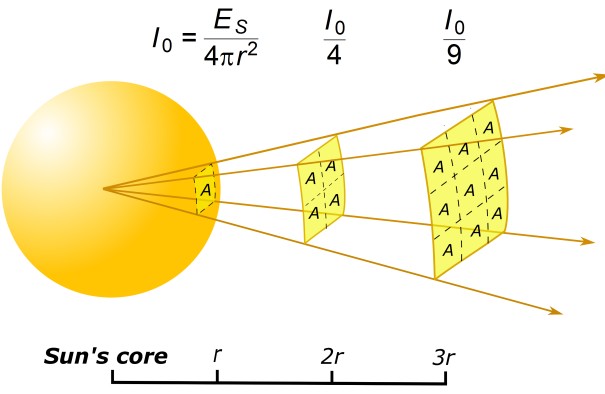

**Figure 3.** Sketch of the solar flux density $(I_0)$ as a function of the distance to the Sun's core.

## 2.3 Modelling the solar irradiance on the top of the atmosphere

In Fu (2006, p. 116), the incoming solar radiation (ISR) on the top of the atmosphere, $S_0$, is estimated by the solar flux density $(I_0)$, which assumes an isotropic concentric emittance from the Sun. The total solar emittance, $E_S$, must be constant, regardless of the size of the sphere area. The flux density adheres to the inverse square law, meaning that energy per area must diminish when the distance from the energy source increases, as shown in Figure 3.

Therefore, $E_S$ can be estimated doing the multiplication of the solar sphere area by their emission in a one-meter square, being the emission calculated using the Stefan-Boltzmann law, $E_b = \sigma T^4$, assuming the photosphere temperature and the solar radius as $T = 6000\,K$ and $r = 6.96x10^8\,m$, respectively Hartmann (2016, p. 29). The flux density on the Earth's orbital position, $S_0$, can be estimated using a hypothetical solar sphere of radius $a = 150x10^9\,m$, the Earth-Sun distance. The solar constant is thus $S_0 \approx 1366\,Wm^{-2}$.

During simplified climate analysis, the global ISR average is $\overline{S_0}$, and because of the large Earth-Sun distance, the solar radiation is assumed to be a parallel and uniform beam Hartmann (2016, p. 31). Thus, to estimate $\overline{S_0}$, the total ISR inside of the disc area, the cross-section, must be divided by the Earth sphere area (Figure 4), resulting in $\overline{S_0} \approx 340\,Wm^{-2}$.

Although simplified climate approaches on zero-dimensional models are viable in classrooms, complex climate investigations require providing a realistic ISR. This way, the DINSOL model implements two different methods to calculate the ISR in the outer atmosphere. The first method is *daily insolation* $(Q_0)$ and the second is *instantaneous solar radiation* $(Q_I)$. Thus, using this information, the methods can now be elaborated on.

### 1) Daily insolation

To calculate the ISR, aspects of ISR incidence on spherical surfaces must be considered (e.g., solar zenith angle ($Z$), hour angle ($H$), solar declination ($\delta$), latitude ($\phi$), and the relative Earth-Sun distance ($\rho$)). Thus, In Liou (2002, p. 51), a realistic model



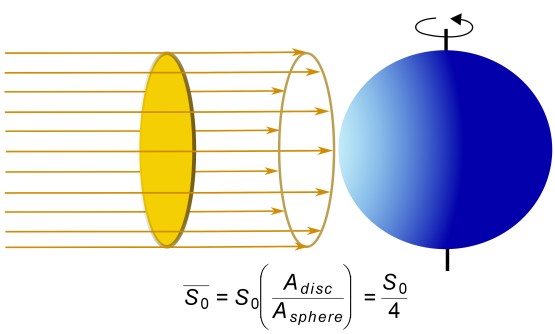

$$\overline{S_0} = S_0 \left( \frac{A_{disc}}{A_{sphere}} \right) = \frac{S_0}{4}$$

**Figure 4.** The incoming solar radiation as a parallel beam and its global average, $\overline{S_0}$.

of the daily insolation can be generated using the equations described below:

$$Q_0 = S_0 \left( \frac{a}{r} \right)^2 \cos Z \tag{7}$$

where $S_0$ is the solar constant and $(a/r)^2$ represents the inverse square of the relative Earth-Sun distance, $1/\rho^2$. The steps

to calculate $\rho$ were listed section 2.1. The solar zenith angle, $Z$, is obtained from the spherical law of cosines, shown in the equation below:

$$\cos Z = \sin \phi \sin \delta + \cos \phi \cos \delta \cos h \tag{8}$$

where $h$ is the hour angle for a small time interval. To obtain the daily insolation, the total ISR at any latitude for a given day requires calculating daily ISR from sunrise ($SR$) to the sunset ($SS$).

$$Q_0 = \int_{SR}^{SS} S_0 \left( \frac{a}{r} \right)^2 (\sin \phi \sin \delta + \cos \phi \cos \delta \cos h) dt \tag{9}$$

In equation 9 the constants for a given day are $\delta$, $\phi$ and $(a/r)^2$. The hour angle and time are associated with the angular speed of Earth, $\Omega$. Thus, a time differential substitution, $dt$, assuming $dt = dh/\Omega$, can be implemented. Result in equation 9 being a function of $h$:

$$Q_0 = \frac{S_0}{\Omega} \left( \frac{a}{r} \right)^2 \left[ \sin \phi \sin \delta \int_{-H}^{H} dh + \cos \phi \cos \delta \int_{-H}^{H} \cos h \, dh \right] \tag{10}$$

where the hour angle between the sunrise until solar noon is $-H$ and from the solar noon until sunset is $+H$, then solving the integral yields equation:

$$Q_0 = \frac{2S_0}{\Omega} \left( \frac{a}{r} \right)^2 (H \sin \phi \sin \delta + \cos \phi \cos \delta \sin H) \tag{11}$$

Therefore, assuming that the angular speed of Earth is $\Omega = 2\pi/86400$, the equation for accumulated solar radiation in one day, that is, the daily insolation can be found. However, the DINSOL model provides the daily average of the ISR, which





requires the simplification of the equation by removing the total seconds per day. Finally, the equation implemented in the DINSOL source code:

$$Q_0 = \frac{S_0}{\pi} \left(\frac{a}{r}\right)^2 (H \sin\phi \sin\delta + \cos\phi \cos\delta \sin H) \tag{12}$$

Now, using Berger (1978b), to estimate $Q_0$ following the equation 12, it is necessary to calculate $\sin\delta$, $\cos\delta$, $\cos H$, $\sin H$, and $H$, requires the implementation of these equations:

$$\sin\delta = \sin\varepsilon \sin\lambda \tag{13}$$

$$\cos\delta = \sqrt[2]{1 - \sin^2\delta} \tag{14}$$

$$\cos H = -\tan\phi \tan\delta \tag{15}$$

where $\lambda$ is the true solar longitude, already explained and $\varepsilon$ is the Earth obliquity, $\cos H$ demands the initialisation of the following conditions:

$$f(H) = \begin{cases} \cos H < -1, & \cos H = -1 \\ \\ \cos H > 1, & \cos H = 1 \end{cases} \tag{16}$$

Finally, these are the last steps:

$$\sin H = \sqrt[2]{1 - \cos^2 H} \tag{17}$$

$$H = \arccos(\cos H) \tag{18}$$

Prior to the initialisation of the *instantaneous solar radiation* method, the equation used to calculate the day length, in terms
of number of hours of sunlight $(N)$ is:

$$N = \frac{2H}{15} \tag{19}$$

**2) Instantaneous solar radiation**

The DINSOL model has a subroutine dedicated to calculating the instantaneous solar radiation, $Q_I$, this subroutine employs equations 7 and 8. The main difference between $Q_I$ and $Q_0$ is that while $Q_0$ stores a single value per day, $Q_I$ can store
several values per day (hours or minutes). Another key difference is that $Q_I$ is calculated globally, while $Q_0$ simulates only the latitudinal effect. Below is the equation 20 used to calculate $Q_I$:

$$Q_I = S_0 \left(\frac{a}{r}\right)^2 (\sin\phi \sin\delta + \cos\phi \cos\delta \cos h) \tag{20}$$

The equation 20 is calculated using four nested loops (Figure 5), where the first loop represents the passage of the days in a year, $D_i$, the second loop is the time interval, $t_i$, within one day (e.g., 6 h, 3 h, 1 h, etc.), such as is that shown





$$
\left\{
\begin{array}{l}
N_d \\
\\
\\
\\
\\
\\
\\
\\
D_i
\end{array}
\right.
\left\{
\begin{array}{l}
nt \\
\\
\\
\\
\\
\\
\\
t_i
\end{array}
\right.
\left\{
\begin{array}{l}
ny \\
\\
\\
\\
\\
\\
y_i
\end{array}
\right.
\left\{
\begin{array}{l}
nx \\
\\
h(t_i, nt, x_i, nx) \\
\\
Q_I(..., h) \\
\\
x_i
\end{array}
\right.
$$

**Figure 5.** An example of the loop employed to run the instantaneous solar radiation subroutine.

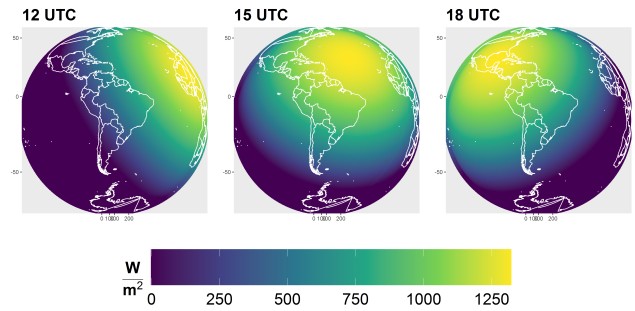

**Figure 6.** The instantaneous solar radiation on the top of the atmosphere from the DINSOL model for nowadays on June 29.

in Figure 6. The third nested loop represents the latitudes, $y_i$, and finally, the fourth contains the longitudinal loop, $x_i$.

Following is given the hour angle, $h$, as a function of the time interval, $t_i$, and longitude $x_i$:

$$
h = -\pi + \left(\frac{2\pi}{nt}\right)(t_i - 1) + \left(\frac{2\pi}{nx}\right)(x_i - 1) \tag{21}
$$

Therefore, after calculating $h$, the results must be substituted into equation 20, where the other variables may be calculated

using the $Q_0$ method, and negative $Q_I$ values are assumed as zero. The algorithm for hour angle initiates from the west in

the eastwards direction, following the Earth's rotation about its axis. Thus, considering the first day, $D_1$, and the first hour (0

UTC), $t_1$, the hour angle must be calculated globally. This means that for each latitude, an hour angle covering all longitudinal

points from the west to the east must be calculated. This way, from the first timestep, $t_1$, the algorithm starts the second

timestep, $t_2$, which is the same data calculated previously at $t_1$, after a westward rotation (Figure 6). Hence, iterations must

repeat for 24 h, before the start of the second day, $D_2$, where changes in solar declination, $\delta$, and Earth-Sun distance, $\rho$, must

be considered. In summary, this subroutine is responsible for realistically simulating the ISR in the outer atmosphere to ensure

that the model is viable for complex climate models or studies that require accurate simulations of the effect of the day-to-night

transition.





**Table 1.** The constant values are used to solve the Berger parameterizations.

|  | $\varepsilon^*$ | $\overline{\psi}$ | $\zeta$ |
|---|---|---|---|
|  | (°) | (arcsec) | (°) |
| Be78 | 23.320556 | 50.439273 | 3.392506 |
| Be90 | 23.3334095 | 50.41726176 | 1.60075265 |

## 2.4 Orbital motions: parameterizations

The law of universal gravitation computes the mutual attraction force between two bodies and, according to Yang (2017), their
formula is:

$$F = -G\frac{Mm}{a^2}\hat{r} \tag{22}$$

for the Sun-Earth case, $M$ represents mass of the sun, $m$ the mass of the earth, $G$ is the universal gravitational constant, $G = 6.6739x10^{-11}Nm^2/kg^2$, and $\hat{r}$ is a unitary vector. The Earth-Sun distance, $a$, is given by the average radius of the Earth's orbit.

Although Newton adopted the gravitational law for two celestial bodies with success (e.g., Sun-Earth, Earth-Moon, Sun-
Mars, etc), three-body problems proved to be more complex. Precise astronomical predictions require considering the grav-
itational influence of the other celestial bodies (Laskar et al., 2004). Euler and Lagrange found particular solutions for the
three-body problem (Musielak and Quarles, 2017). In the late 1880s, Heinrich Bruns and Henri Poincaré showed that a general
arrangement of three or more bodies (*n-body problem*) yielded no analytical solution (Hamilton, 2016).

Though complex astronomical motions cannot admit analytical solutions, a distinguished researcher overcame this obstacle
by adopting the spectral decomposition technique. André Berger made it possible for anyone to estimate the Earth's orbital pa-
rameters (EOP) within ± 1 Myr (Crucifix et al., 2009). His parameterization was published firstly in Berger (1978b), and after
some years, a newer version in Berger and Loutre (1991) expanded the valid time range for ± 3 Myr. Now, in this chapter and
the following, Berger (1978b) and Berger and Loutre (1991) will be referred to as Be78 and Be90, respectively. Both parameter-
izations, Be78 and Be90, are described in this section with the focus on the main formulas and tables presented in this classical
paper.

Another remarkable researcher, Jacques Laskar, also simulated new long-term solutions for EOP. Berger used the Laskar
solutions in Be90 (Laskar, 1986, 1988). Furthermore, within the last decades, Laskar published novel solutions focused on
expanding the valid time range (e.g., Laskar et al., 1993, 2004, 2011). Further, though current computers can simulate the
planetary motions around the Sun for billions of years, the chaotic behaviour of the solution still limits the validation to a few
tens of millions of years (Laskar et al., 2011).

Thus, the efforts of André Berger and Jacques Laskar represent an important contribution to paleoclimatology. This way,
following the idea of this chapter, the mathematical description of the Berger parameterizations, covering from equations to
constants will be provided, as well as a Laskar custom parameterization, such as in the DINSOL source code.





### 1) The analytical solution of Be78 and Be90

Both parameterizations, Be78 and Be90, use spectral analysis, which used the same trigonometrical equations. However, Be78 and Be90 methods require distinctive data sources to work. Thus, DINSOL contains three tables for every Berger parameterization: obliquity, eccentricity, and precession, such as tables 1, 4, and 5 in Be78, respectively. Furthermore, all the data were obtained from PALINSOL, an R package, and converted to sequential binary files. The data contained the following columns: amplitude, mean rate, phase, and period. The time in years for Be78 or Be90 parameterizations is represented in the equations 260 by the $t$ variable.

The calculation of the obliquity ($\varepsilon$) is then estimated with the equation:

$$\varepsilon = \varepsilon^* + \sum_{i=1}^{N} A_i \cos\left(R_i t + F_i\right) \tag{23}$$

where $\varepsilon^*$ is a constant given in table 1, $N$ representing the number of the terms per column, and $A_i$, $R_i$, and $F_i$ are, respectively: the second, third, and fourth columns of the obliquity tables used in Be78 and Be90. Similarly, to the obliquity, the 265 eccentricity, $e$, can be calculated from the following equations:

$$e \sin \pi = \sum_{i=1}^{N} M_i \sin\left(G_i t + B_i\right) \tag{24}$$

$$e \cos \pi = \sum_{i=1}^{N} M_i \cos\left(G_i t + B_i\right) \tag{25}$$

$$e = \sqrt{(e \sin \pi)^2 + (e \cos \pi)^2} \tag{26}$$

where $M_i$, $G_i$, and $B_i$ are, respectively: the second, third, and fourth columns of the eccentricity tables used in Be78 and Be90, and $N$ is the number of elements per column. To calculate the precession, $\varpi$, the following these steps must be followed:

$$\psi = \overline{\psi} t + \zeta + \sum_{i=1}^{N} P_i \sin\left(K_i t + D_i\right) \tag{27}$$

being $\psi$ the general precession, and $\overline{\psi}$ and $\zeta$ constants are given from table 1, as well as $P_i$, $K_i$, and $D_i$ are, respectively: the 275 second, third, and fourth columns of the precession tables used in Be78 and Be90. Moreover, $N$ is the number of the table terms. Now, we also need to calculate the arctangent from $e \sin \pi$ and $e \cos \pi$, like the equation below:

$$\arctan \pi = \arctan \left( \frac{e \sin \pi}{e \cos \pi} \right) \tag{28}$$

if $\arctan \pi$ is negative, 180° must be added for $\arctan \pi$ or else it must be kept unchanged. The longitude of the perihelion, $\varpi$, can now be calculated using the equation:

$$\varpi = \arctan \pi + \psi + \pi \tag{29}$$





If $\varpi$ is greater than 360°, a subtraction: $\varpi - 360°$, must be performed to obtain an angle less than the length of a full revolution. The longitude of perigee, $\omega$, can be calculated as $\omega = \varpi + 180°$. Additionally, it is crucial use the same units: radians, degrees, and arcseconds. For instance, DINSOL Berger subroutines convert all column data to radians.

**2) The Laskar solutions**

The DINSOL model combines two Laskar time series solutions, Laskar et al. (2004, 2011), resulting in a time range from -249 Myr to 21 Myr, referred to as the Lask. It implements the function $s(y)$, where $y$ is a year multiple of thousand. However, if $y$ is not a multiple of thousand, $s(y)$ cannot be used directly and a simple slope-intercept equation must be used where the two nearest points are defined as: before $(t_i)$ and after $(t_{i+1})$. For instance, when $y = -4600$, then $t_i = -5000$ and $t_{i+1} = -4000$. Thus, the effective time, $t$, used in the slope-intercept equation is $t = y - t_i$, which results in a time of $t = 400$.

Below is the Equation 30, where $f(y)$ may assume two forms:

$$f(y) = \begin{cases} s(y), & y \bmod 10^3 = 0 \\ \\ \theta t + b, & y \bmod 10^3 \neq 0 \end{cases} \tag{30}$$

been $b = s(t_i)$ and slope given by:

$$\theta = \frac{s(t_{i+1}) - s(t_i)}{10^3} \tag{31}$$

The annual change for each EOP is small, resulting in a small calculation error from the slope-interception equation. Additionally, when the Be78 and Lask data calculated from DINSOL were compared, Figure 7 showed that the graphs overlap; therefore, the Lask parameterization can accurately estimate the nonexistent Laskar original data. Both Laskar solutions were obtained from the web page (http://vo.imcce.fr/insola/earth/online/earth/earth.html, last access: 2 August 2022) that is maintained by the Institut de Mécanique Céleste et de Calcul des Ephémérides (IMCCE).



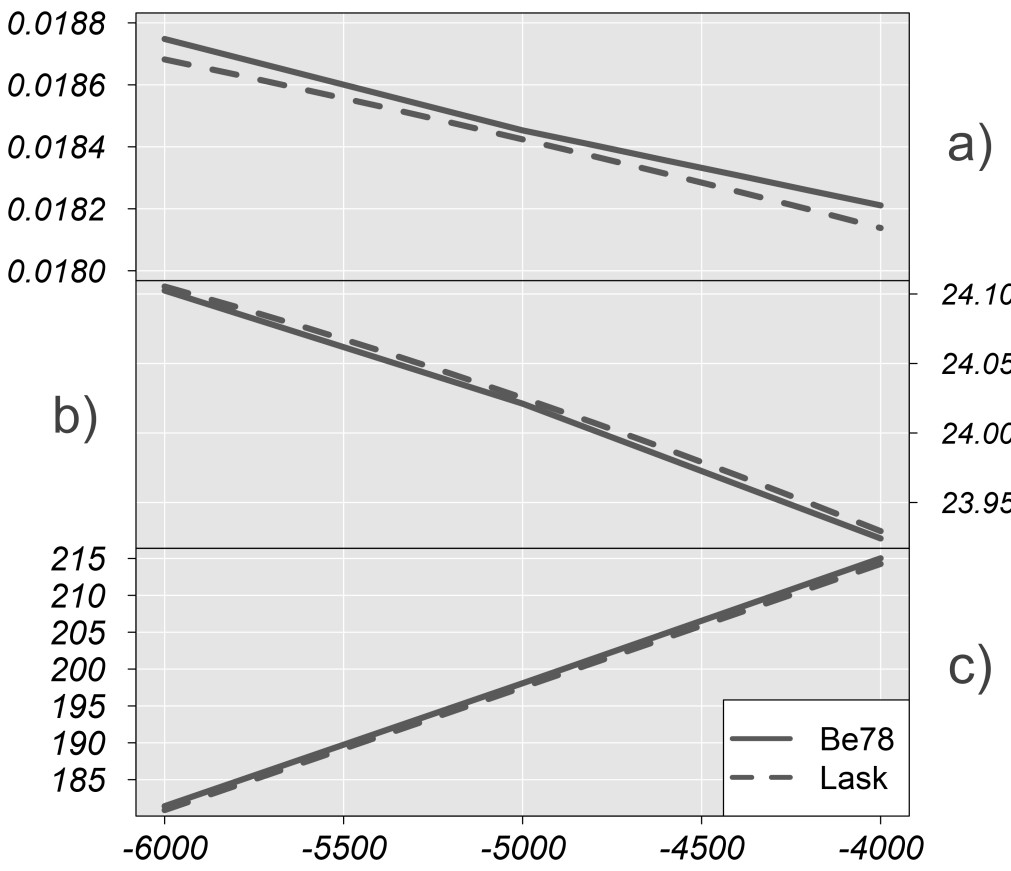

**Figure 7.** Comparison between Be78 and Lask data, with time (in years) on the horizontal axis. Additionally, graph (a) is eccentricity, (b) is obliquity [°], and (c) is precession [°].





## 2.5 Model structure, setup, and execution

The DINSOL source code has a simple hierarchical structure written in Fortran. A flow diagram representing the DINSOL source code is available in Figure 8, including input data, the main program, modules, functions, subroutines, and output data. The model initiates from the main program reading the input data: tables for analytical solutions of Be78 and Be90 and combined time series of Lask. The program also read data from the namelist file (Table 2) as well as a module containing the declared variables and simulation parameters. In the next step, the program invokes the DINSOL subroutine, which is the main

function and calls on all other functions for the simulations.

The next steps are a series of commands executed from the namelist setup function, where the calendar type must be initialised. Variables like spatial resolution, time interval, and calendar variables are then declared. The next step invokes parameterization to obtain the orbital parameters, and then the subroutines (seasons, perigee, and apogee) implement a calendar function to determine the day and month occurrences of summer, autumn, winter, perihelion, and aphelion. The solar longitude

subroutine is then called, and it computes the annual true solar longitude for use in the daily insolation and instantaneous irradiance subroutines. Finally, the output subroutine is responsible for storing the data simulated during the DINSOL execution.

The DINSOL model also works from a Graphical User Interface (GUI), where the namelist options are the same, except for the spatial resolution, where the GUI mode offers only six options: 5°, 4°, 3°, 2°, 1°, and 0.5°. All the output files generated (Table 3) are identical regardless of the execution mode, except for the plot file *gui-plot.png*: a panel used to display results in

the GUI mode. This plot contains sketches of the orbital parameters and contour fields for a simulated year: daily insolation, the difference to current days, and the length of a day. A snapshot of the GUI interface and two graphical windows containing the results are displayed in Figure 9.

Users have the option to customise their simulation setup, implementing custom scenarios using *user-defined value*. The orbital parameters can then be set without compliance to the Berger or Laskar parameterizations, provided that the results will

not be invalidated. For instance, if the eccentricity is set to zero, the orbit of Earth would become a perfect circle, meaning the dates of perihelion and aphelion would no longer exist. If obliquity is set to zero, the seasons would not exist, and if a negative obliquity is assumed, the solstice and equinox dates would occur on inverted dates. Even though assume hypothetical cases, the program still works correctly.



**Table 2.** A short description of DINSOL namelist variables.

| Variables | Short description |
|---|---|
| YEAR | Any integer number in the time-slice: -249 Myr to 21 Myr. <br> *Note: zero is representing nowadays.* |
| S0 | The solar constant in the range $]0 : 10^8[$ . <br> *Note: S0 must be in $W/m^2$* |
| NY | Latitudinal number points. |
| NX | Longitudinal number points. |
| NTIME | The time interval within one day, given in hours or minutes: <br> 1 - 6 hours <br> 2 - 3 hours <br> 3 - 1 hour <br> 4 - 30 minutes <br> 5 - 15 minutes |
| CALENDAR | Define the number of days in the year: <br> 1 - 365 days <br> 2 - 360 days |
| ORBITAL | Define the orbital parameterization: <br> 1 - Be78: $\pm$ 1 Myr. <br> 2 - Be90: $\pm$ 3 Myr. <br> 3 - Lask: -249 Myr to 21 Myr. <br> 4 - User-defined value. |
| ECC | Eccentricity: any value in the range [0:0.5]. |
| OBLQ | Obliquity: any value in the range [-90:90]. |
| PRCS | Precession: any value in the range [0:360[. <br> *Note: This is the longitude of the perihelion, $\varpi$.* |





**Table 3.** A short description of DINSOL output data.

| Files | Short description |
|---|---|
| summary.txt | This file is a resume of the main results: |
| | - Namelist setup |
| | - The orbital parameters |
| | - Astronomical dates |
| | - Annual average of daily insolation |
| insolation.txt | A formatted data containing some results in columns: |
| | - Year |
| | - Days |
| | - True solar longitude ($\lambda$) |
| | - Relative Earth-sun distance ($\rho$) |
| | - Latitude ($\phi$) |
| | - Solar declination ($\delta$) |
| | - Day length ($N$) |
| | - Daily insolation ($Q_0$) |
| solar.radiation | A binary file of annual daily insolation ($Q_0$). |
| solar.radiation.ctl | A GrADS descriptor file. |
| radiation | A binary file of instantaneous solar radiation ($Q_I$). |
| radiation.ctl | A GrADS descriptor file. |







**Figure 8.** Fortran representation of the DINSOL source code, with subroutines, modules, namelist, input and output data and brief explanations of the simulation steps.





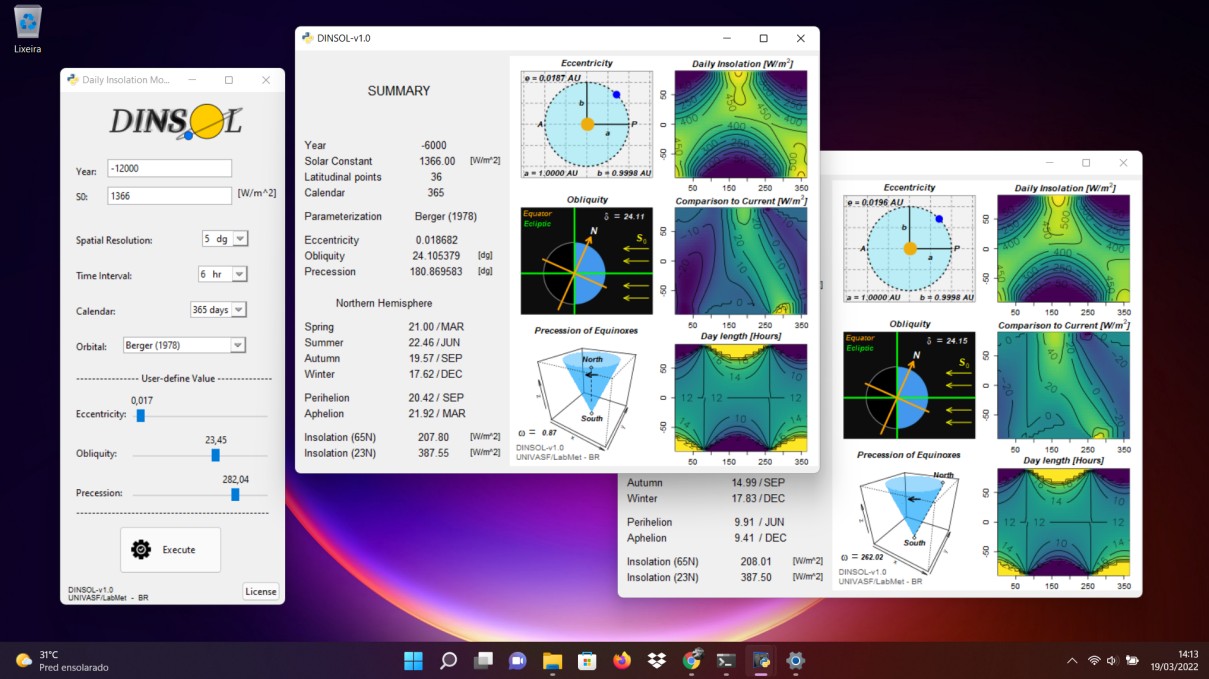

**Figure 9.** Snapshot of DINSOL-v1.0 GUI running on Windows 11.

## 3   Model evaluation: a short statistical analyse

In this chapter, the model was evaluated by analysing the confidence level of the results obtained from the model. The first subchapter contains the Earth's orbital parameters (EOP) evaluation, where the orbital parameterizations (Be78, Be90, Lask) are compared with Bartlein and Shafer (2019) EOP data. The authors adopted the Goddard Institute for Space Studies (GISS) Fortran programs to run the EOP using Be78 parameterization. The time series' start from -150 kyr until the present day, $t = 0$. Furthermore, we decided to use the climatic precession ($e \sin \varpi$) instead of the distance of perihelion, $\varpi$.

In the following subchapters, standard products were analysed using comparisons between PMIPII and DINSOL. At first, the astronomical dates were evaluated, where data for different periods was analysed by comparing DINSOL with PMIPII dates. Please note that the classical method of measuring the paleoclimate dates assumes the spring equinox as a fixed date (March 21). In the final subchapter, the monthly insolation data evaluations were conducted where the assumption was that the solar constant was $S_0 = 1365 \, W m^{-2}$; year equal to zero for current days (0K); $t = -6000$ to mid-Holocene (6K); and $t = -21000$

to last glacial maximum (21K); using a 365-day and 360-day calendar with the Berger (1978b) parameterizations (EOP and daily insolation) as in the PMIPII experiments implemented.

The samples were analysed from measures of central tendency using average ($\bar{X}$) and median ($\mu$), also measures of dispersion using standard deviation ($\sigma$) and coefficient of variation ($C_V$), as well as inferential statistical analysis. The Root Mean





Square Error (RMSE) was implemented to evaluate the sample differences, where the RMSE was calculated by performing the
sum of the differences between simulation and observational data, given by equation 32:

$$RMSE = \sqrt[2]{\frac{1}{N}\sum_{i=1}^{N}(S_i - O_i)^2} \tag{32}$$

where conform Oliveira et al. (2019) $N$ is the number of elements, $S_i$ representing the simulation data while $O_i$ is observational
data, noting that PMIPII and GISS have been considered as observational data, a standard dataset.

Inferential statistical analysis was also performed, and after performing the normality tests, the conclusion was that the
samples did not exhibit a normal distribution. The consequent step was to conduct the Wilcoxon test (U test) which is a non-
parametric hypothesis test. This test makes a Rank-Sum Test with continuity correction, comparing two sample medians (Lin
et al., 2021). This test is typically referred to as the Mann-Whitney test or alternatively, the U test and it is an alternative to the
t-test because it does not require as many assumptions about the sample. Therefore, the U test was used in conjunction to the
hypothesis below:

$$H_0 : \mu_D = 0$$

$$H_1 : \mu_D \neq 0$$

where the null hypothesis, $H_0$, assumed that samples median differences, $\mu_D$, must be zero. While the alternative hypothesis, $H_1$,
considered that samples were offset. The U test was implemented with a standard significance level, $\alpha = 0.05$.

### 3.1 Earth's orbital parameters

In Figure 10, there is a correlation between the GISS and DINSOL curves, indicating that DINSOL EOP subroutines (Be78,
Be90, and Lask) function as expected. The error margin estimations were determined using a statistical summary shown in
Table 4. Thus, measurements such as mean, $\bar{X}$, median, $\mu$, standard deviation, $\sigma$, and coefficient of variation, $C_V$, indicate a
strong correlation between DINSOL and GISS. Therefore, suggesting an insignificant difference between DINSOL and GISS
samples. Additionally, this may be validated from the U test, where the *p-value* is greater than the standard significance level,
$\alpha = 0.05$, for all sample analyses, implying that the median differences, $\mu_D$, may be zero. The RMSE values also support the
previous statistical inference from the insignificant difference between the samples.

The GISS and Be78 values are nearly identical, which validates the Be78 subroutine. The same occurs with Be90 and Lask
parameterizations, although the differences are significant, it is worth mentioning that each Berger parameterization is only
valid for specific time intervals. Nevertheless, analysing only the past 150 kyr, the conclusion is that the EOP subroutines
in the DINSOL source code work conform to expectations. The numerical differences are affected by the epoch and ecliptic
references used by each parameterization, that is, the time reference. Thus, while Be78 assumed 1950 as the epoch and 1850 as
its ecliptic reference, Lask parameterization adopted J2000 as a time reference, conforming to the Julian calendar. Therefore,
when the same time was chosen in each parameterization, what was calculated was the EOP for different epochs. Consequently,
Be78 and Lask parametrizations require a 50-year calibration shift.





**Figure 10.** Time series of the Earth's orbital parameters (EOP) for the last 150 kyr. a) Eccentricity, b) Obliquity and c) Climatic precession. Each graph brings Goddard Institute for Space Studies (GISS) and DINSOL parameterizations: Be78, Be90, and Lask.





**Table 4.** Evaluation of the DINSOL model adopting a statistical analysis from Earth's Orbital Parameters (EOP), where the values were from minima each calendar type (*Min*), the maxima (*Max*), average ($\bar{X}$), median ($\mu$), standard deviation ($\sigma$), coefficient of variation ($C_V$), the root mean square error (*RMSE*), and the *p-value* from the Wilcoxon test (*U test*).

| | *Min* | *Max* | $\bar{X}$ | $\mu$ | $\sigma$ | $C_V$ | *RMSE* | *U test* *p-value* |
|---|---|---|---|---|---|---|---|---|
| GISS [$e$] | 0.012509 | 0.041421 | 0.026755 | 0.026859 | 0.010076 | 0.37660 | - | - |
| Be78 [$e$] | 0.012509 | 0.041421 | 0.026755 | 0.026859 | 0.010076 | 0.37660 | 7.3008E-09 | *ns* |
| Be90 [$e$] | 0.014093 | 0.043988 | 0.027717 | 0.026111 | 0.010429 | 0.37628 | 0.0013568 | *ns* |
| Lask [$e$] | 0.013706 | 0.043921 | 0.027408 | 0.025952 | 0.010528 | 0.38411 | 0.0011497 | *ns* |
| GISS [$\varepsilon$] | 22.20748 | 24.43585 | 23.35631 | 23.44278 | 0.711554 | 0.03047 | - | - |
| Be78 [$\varepsilon$] | 22.20748 | 24.43585 | 23.35631 | 23.44278 | 0.711554 | 0.03047 | 1.3622E-06 | *ns* |
| Be90 [$\varepsilon$] | 22.23484 | 24.43667 | 23.36954 | 23.43276 | 0.702570 | 0.03007 | 0.0205545 | *ns* |
| Lask [$\varepsilon$] | 22.20721 | 24.43464 | 23.35852 | 23.43071 | 0.709339 | 0.03037 | 0.0281573 | *ns* |
| GISS [$e\sin\varpi$] | -0.0413 | 0.039898 | -0.000147 | -0.000187 | 0.020279 | -137.686 | - | - |
| Be78 [$e\sin\varpi$] | -0.0413 | 0.039898 | -0.000147 | -0.000187 | 0.020279 | -137.682 | 3.0940E-08 | *ns* |
| Be90 [$e\sin\varpi$] | -0.0439 | 0.041798 | -0.000131 | -0.000057 | 0.021009 | -160.797 | 0.001249 | *ns* |
| Lask [$e\sin\varpi$] | -0.0439 | 0.041907 | -0.000132 | -0.000386 | 0.020825 | -158.399 | 0.001075 | *ns* |

*ns : non-significant*





**Table 5.** This table contains the dates of summer, autumn, winter, perihelion, and aphelion during the present (0K), mid-Holocene (6K), and last glacial maximum (21K) determined using DINSOL and PMIPII for 365-day and 360-day calendars.

|  |  | Summer solstice (*June*) | Autumnal equinox (*September*) | Winter solstice (*December*) | Perihelion | Aphelion |
|---|---|---|---|---|---|---|
| 0K | PMIPII [365] | 21.73 | 23.30 | 22.05 | 2.85/01 | 4.35/07 |
|  | DINSOL [365] | 21.74 | 23.30 | 22.05 | 2.85/01 | 4.35/07 |
|  | PMIPII [360] | 22.46 | 24.74 | 23.26 | 4.91/01 | 4.91/07 |
|  | DINSOL [360] | 22.47 | 24.76 | 23.28 | 4.91/01 | 4.91/07 |
| 6K | PMIPII [365] | 22.45 | 19.56 | 17.61 | 20.42/09 | 21.92/03 |
|  | DINSOL [365] | 22.46 | 19.57 | 17.62 | 20.42/09 | 21.92/03 |
|  | PMIPII [360] | 23.17 | 21.06 | 18.89 | 21.90/09 | 21.90/03 |
|  | DINSOL [360] | 23.18 | 21.07 | 18.91 | 21.90/09 | 21.90/03 |
| 21K | PMIPII [365] | 21.32 | 23.52 | 22.65 | 15.51/01 | 17.01/07 |
|  | DINSOL [365] | 21.32 | 23.52 | 22.65 | 15.51/01 | 17.01/07 |
|  | PMIPII [360] | 22.06 | 24.96 | 23.86 | 17.39/01 | 17.39/07 |
|  | DINSOL [360] | 22.06 | 24.97 | 23.87 | 17.39/01 | 17.39/07 |

## 370  3.2  Astronomical dates

Table 5 contains the dates of summer, autumn, winter, perihelion, and aphelion simulated second PMIPII and DINSOL. These dates represent the current days (0K), mid-Holocene (6K), and last glacial maximum (21K). Thus, perihelion and aphelion dates calculated by DINSOL are identical to those calculated using PMIPII. These dates correlate with the expected dates because they were determined using equations 4, 5, and 6. Nevertheless, the summer, autumn, and winter dates have a small

error to the PMIPII-determined. In the DINSOL, these dates were estimated from a large number of values within the annual true solar longitude vector. In table 5, an accumulative error is observed, where the season date error increases as a function of the distance to the spring equinox. The RMSE provides values around 12, 16, and 19 min for summer, autumn, and winter, respectively. Thus, for lesser than 20 min, the error remains small, meaning that the DINSOL provides accurate date estimates. The compliance of DINSOL and PMIPII to the classical method of measuring astronomical events must always be considered.

However, for a more realistic approach to paleoclimate calendars, it is recommended to follow the methodology of Joussaume and Braconnot (1997) or Bartlein and Shafer (2019), although many authors prefer the classical method because it is easier to compare with our current Gregorian calendar.





### 3.3 Monthly insolation

The monthly insolation was analysed from a visual comparison of the contour fields (Figure 11) and data from a statistical
approach (Table 6). The evaluation uses PMIPII insolation samples prepared from 365-day and 360-day calendars at three
different periods: present (0K), mid-Holocene (6K-0K), and last glacial maximum (21K-0K). Simulations were thus performed
with DINSOL using the same PMIPII setup. Here, the contour fields in Figure 11 (a - f) were generated using a 365-day
calendar setup, with (g - l) employing a 360-day calendar setup. From Figure 11, compared to the current, the major insolation
differences were found during the mid-Holocene. The perihelion date changes affected the incoming solar radiation more than
other orbital parameters during the mid-Holocene.

Nevertheless, the image sets exhibit similarities in the contour fields of the DINSOL (Figures 11a, 11b, and 11c) and PMIPII
(Figures 11d, 11e, and 11f). This means that the daily insolation subroutine present in the DINSOL is viable from its correlation
to statistical data in Table 6. Therefore, from the range of the data, the minima and maxima are nearly identical and converge
towards the same value in the 0K-6K range. Furthermore, the central tendency indicates that average measurements are nearly
identical, with the median values being identical for all. The dispersion values, exhibit the previously observed behaviour,
validating the previous conclusions. The standard deviations of DINSOL and PMIPII represent the same data. Please note that
the RMSE provides a very small difference between the samples, where the RMSE average is about $0.013\ Wm^{-2}$. Finally, the
U test confirms our inference, meaning that there are no significant differences between the DINSOL and PMIPII by consider
samples for a 365-day calendar.

Now, focusing on a 360-day calendar, where we have the DINSOL minus PMIPII (Figure 11g, 11h, and 11i) and the monthly
insolation at the top of the atmosphere for 0K, 6K, and 21K (Figure 11j, 11k, and 11l). The field differences show us almost
homogenous contour data with values near zero. Indeed, according to the Table 6, all samples must be considered numerically
the same because we did not find significant differences in our statistical analyses considering a 360-day calendar.



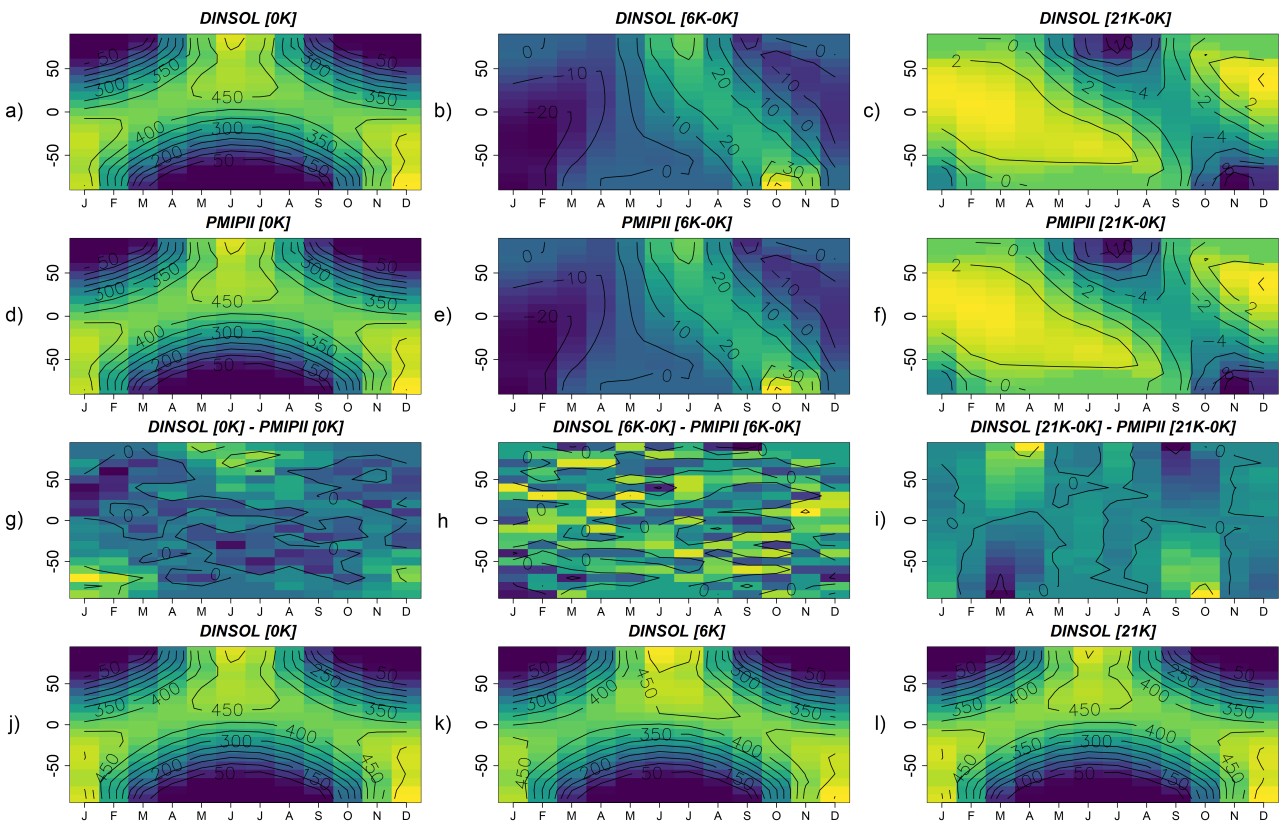

**Figure 11.** Monthly insolation contour fields obtained from a 365-day calendar: a) Current days from DINSOL, b) mid-Holocene minus current from DINSOL, c) Last maximum glacial minus current from DINSOL, d) Current days from PMIPII, e) mid-Holocene minus current from PMIPII, and f) last maximum glacial minus current from PMIPII with the contour fields obtained from a 360-day calendar, g) DINSOL minus PMIPII to current days, h) DINSOL minus PMIPII to mid-Holocene, i) DINSOL minus PMIPII to last maximum glacial, J) Current days from DINSOL, k) mid-Holocene from DINSOL, and l) Last maximum glacial from DINSOL. The horizontal axis represents the months while the vertical axis is the latitude.





**Table 6.** Evaluation of the DINSOL model adopting a statistical analysis from monthly insolation data samples, where the data displayed represents the minimum values of each data (*Min*), the maximum values (*Max*), average ($\bar{X}$), median ($\mu$), standard deviation ($\sigma$), coefficient of variation ($C_V$), the root mean square error (*RMSE*), and the *p-value* from the Wilcoxon test (*U test*).

| | | *Min* | *Max* | $\bar{X}$ | $\mu$ | $\sigma$ | $C_V$ | *RMSE* | *U test* *p-value* |
|---|---|---|---|---|---|---|---|---|---|
| 0K | DINSOL [365] | 0 | 550.39 | 292.58 | 341.62 | 169.3383 | 0.5788 | 0.02595 | *ns* |
| | PMIPII [365] | 0 | 550.40 | 292.58 | 341.60 | 169.3369 | 0.5788 | | |
| | DINSOL [360] | 0 | 548.26 | 292.62 | 341.41 | 169.4682 | 0.5791 | 0.00481 | *ns* |
| | PMIPII [360] | 0 | 548.25 | 292.62 | 341.41 | 169.4666 | 0.5791 | | |
| 6K-0K | DINSOL [365] | -25.50 | 52.14 | 1.0466 | -0.081 | 16.0434 | 15.3283 | 0.00313 | *ns* |
| | PMIPII [365] | -25.50 | 52.14 | 1.0466 | -0.085 | 16.0435 | 15.3288 | | |
| | DINSOL [360] | -25.61 | 51.83 | 1.15 | -0.15 | 16.0347 | 13.9873 | 0.00290 | *ns* |
| | PMIPII [360] | -25.61 | 51.83 | 1.15 | -0.16 | 16.0347 | 13.9854 | | |
| 21K-0K | DINSOL [365] | -13.56 | 4.16 | -0.83 | 0 | 4.0808 | -4.9441 | 0.01966 | *ns* |
| | PMIPII [365] | -13.54 | 4.15 | -0.83 | 0 | 4.0812 | -4.9456 | | |
| | DINSOL [360] | -13.48 | 4.16 | -0.84 | 0 | 4.0872 | -4.8422 | 0.02012 | *ns* |
| | PMIPII [360] | -13.46 | 4.15 | -0.84 | 0 | 4.0876 | -4.8424 | | |

*ns : non-significant*





# 4 Conclusions

The Daily Insolation (DINSOL-v1.0) model is a robust and versatile tool offering various simulation options and applications. It is ideal for gathering solar radiation data in climate models, which aids in simplified climate approaches or more complex studies. Furthermore, the DINSOL model may be employed as an educational tool when incorporated with a user-friendly graphical user interface (GUI). The program was developed as a novel educational tool for use in geosciences colleges worldwide, providing a quick view of daily insolation by considering the Milankovitch cycles (ideal for paleoclimatology)

or adopting hypothetical orbital parameters (ideal for exoplanets). Therefore, the program can aid students, teachers, and researchers in conducting their respective studies. The DINSOL is an open-source Fortran 90 program, encouraging its users to continuously develop the source code, with its GUI written in PyGTK, and graphic results plotted using a custom R script. Despite the model evaluation presenting insignificant differences to the models in published literature, users are incentivised to report issues when running the program. The DINSOL program may be considered a viable tool that provides various types

of output data and realistically simulates the instantaneous solar radiation globally, simulating the passage of day to night.

*Code and data availability.* The DINSOL program is available online from the Zenodo repository (https://doi.org/10.5281/zenodo.6884499, last access: 22 July 2022), also on GitHub (https://github.com/Emerson-D-Oliveira/dinsol-v1.0, last access: 22 July 2022) as well as in the Laboratory of Meteorology (LabMet) web page (http://labmet.univasf.edu.br/joomla/index.php/pesquisas/modelos-numericos, last access: 22 July 2022). The versions available in Zenodo and GitHub are the same files, while the LabMet contains three different versions. Where the

LabMet page provides a standard version, the same as Zenodo and GitHub, and still two custom versions created to facilitate the installation process in Ubuntu and Windows operating systems. The custom versions contain a user manual explaining step-by-step the installation and program execution process. It is important to mention that the Windows version does not require any prior programming experience from the user for the installation of the program and execution in graphical mode. The versions available in the Zenodo and GitHub repositories have a README file explaining how to install and execute the DINSOL. The standard version requires the users to have some experience

with Linux distros, which is necessary to install the libraries individually. Although a macOS version of the program is not available, users are encouraged to install the DINSOL program on different operating systems.

*Sample availability.* A supplementary file containing all datasets used to prepare this manuscript is available in the Zenodo repository (https://doi.org/10.5281/zenodo.6885502, last access: 22 July 2022).

# Appendix A: Data manipulation

The output folder contains five scripts to aid users in analysing the simulation data. The files are listed with a brief description in Table A1.





**Table A1.** Brief description of all scripts located in the DINSOL output folder.

| Scripts | Short description |
|---|---|
| get-dinsol-value.R | This script provides the insolation and day length from a given day and latitude. Moreover, this script still plots a panel containing six graphs. |
| plot-dinsol.R | This script makes a plot with contour fields of the daily insolation. |
| plot-dinsol.gs | This script makes a plot with contour fields of the daily insolation. |
| plot-global.R | This script makes a plot of the instantaneous solar radiation globally from a given day and hour. |
| plot-radiation.gs | This last script makes an animation using the GrADS graphical window. |

## Appendix B: Computing power

The DINSOL does not adopt restrictions for latitudinal and longitudinal number points during execution by command lines.
However, instantaneous solar radiation is recorded in the file *radiation*, and this file is built from four nested loops, as presented

in chapter 2.3. Therefore, the file size can become large when the simulation adopts a high-resolution setup. For example, adopting a spatial resolution of 0.125° (NY=1440 and NX=2880), a 365-day calendar year, and a set time interval of 30 min. The associated file size is approximately 280 GB. Thus, it is necessary to ensure that their computer has sufficient storage capacity in its hard disk when prior to conducting a high-resolution simulation; otherwise, their system could crash, interrupting the simulation.

In contrast, adopting the same simulation setup described above, we should get an output file *solar.radiation* of 2 MB because it records only one data per day for each latitude. In comparison, the *insolation.txt* achieves 46 MB because it contains more variables. Please note that when the *radiation* file is not required, the simulation parameters are set to NX=1 and NTIME=1, allowing users to change the values of NY while keeping simulation times minimal. The most efficient option would be to modify the source code, deactivating the instantaneous solar radiation subroutine and not storing the *radiation* data.





*Author contributions.* Oliveira E.D wrote independently all DINSOL source code, scripts, manuals, as well as this manuscript.

*Competing interests.* The author declares that have no conflict of interest.

*Acknowledgements.* We thank Michel Crucifix, André Berger, and Jacques Laskar for providing the input data used in DINSOL parameterizations. Moreover, we are grateful to Patrick Bartlein for providing the time series of the Earth's orbital parameters and the Paleoclimate Modelling Intercomparison Project (PMIP) team for providing the monthly insolation data. Both data were essential to the model evaluation
process. Furthermore, we thank the Federal University of Vale do São Francisco (UNIVASF) and the Laboratory of Meteorology (LabMet) for incentivising the development of the DINSOL model. Finally, we thank the commentaries and suggestions given by reviewers and editors.



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
