# Peer review of "Daily INSOLation (DINSOL-v1.0): an intuitive tool for classrooms and specifying solar radiation boundary conditions"

_Geoscientific Model Development, 2022_

## Author Comment (AC1)

**Community feedback**

Kevin Schwarzwald

**General Comments**

The author introduces a new modeling package to create values for incoming top of atmosphere solar radiation based on variable orbital parameters and insolation constants. The purpose of the package is to easily create useable inputs for earth system modeling and educational purposes.

The paper details the construction of the model and compares output to existing PMIPII simulations. It seems like the fundamental calculations done by the model are thoroughly detailed, and the model seems to compare well to the state-of-theart simulations in PMIP. I think the paper could however benefit from some clarifications and figures could be made more readable (both detailed below). In particular, DINSOL's advantages over other TOA shortwave calculators could be made clearer.

I was able to easily download and run the model on a Linux server. I was unable to test the GUI, however – I run MacOS (which the paper clarifies is not supported by the software), and the Linux server I have access to runs CentOS, which has known issues with installing required dependencies such as PyGTK.

Since the paper emphasizes the educational component of the model, moving forward (beyond the publication in this journal), it may be helpful to bundle the code with a sample lesson plan to detail how it can be used in the classroom.

**Specific Comments**

L55: This paragraph would benefit from a clearer description of what benefits DINSOL has over existing programs. Though a sentence is given for this point, I was still a bit confused as to these differences – is it usability? Speed? Flexibility?

Author answer:

The most significant DINSOL advantage is the flexibility (offering some options to prepare custom datasets) and usability (ideal for assisting research or being employed in classrooms).

**L129: Most GCMs use 365-day years these days – though this might not be the case for other types of models, perhaps a clarification could help**

Author answer:

The audience will likely employ the DINSOL on simplified and intermediatecomplexity climate models. It's expected that this audience needs a 360-day calendar, such as the Planet Simulator (PlaSim) model (link: https://www.mi.unihamburg.de/en/arbeitsgruppen/theoretische-meteorologie/modelle/plasim.html).

For instance, the DINSOL source code was adapted by myself to work with the Global Resolved Energy Balance (GREB) model, a simplified climate model that works with a 365-day calendar (two-time steps per day).

The main idea is to cater to the users with as many options as possible.

**L335 – 354: Please clarify what exactly is meant by 'sample' – samples of eccentricity, etc.? or Q?**

Author answer:

The samples are referred to the Earth's orbital parameters (EOP), calendar dates, and monthly daily Insolation (Q).

I should be clearer in the paragraph. I will improve it.

**L355: Please clarify which time intervals**

Author answer:

Thanks for warning me. I'm going to introduce the time slice at the initial lines.

**L400: why were the differences between DINSOL and PMIPII only with the 360-day calendar?**

Author answer:

A full analysis of the DINSOL and PMIPII monthly insolation data is given in Table 6, where the Root Mean Square Error (RMSE) and U test were adopted, and conforming to Table 6, we know that DINSOL samples do not differ significantly from the PMIPII. Thus, Figure 11 provides just an overview of these data, where the first six graphs represent the monthly insolation from a 365-day calendar, whereas the rest represent a 360-day calendar.

Appendix B: - this is useful performance information, thanks for including it. Could you also include an example for a more 'typical' GCM-level output (say, 1 degree x 1 degree, while keeping the 30-minute timestep)

Author answer:

I will modify the text in order to include a more typical GCM's spatial resolution (e.g., 1°x1°; 0.5°x0.5°; 0.25°x0.25°) and keep the 30-minutes timesteps. A table with this information should be an interesting alternative as well.

**Typos / Other fixes**

Author answer:

The text typos corrections and other sentence improvements are ongoing.

A few places, including L80, L158, L233, could you please use \$\times\$ instead of \$x\$ for the multiplication sign? The latter makes it look at first glance as if a variable is being referred to.

L92: please clarify true anomaly of what (solar longitude I assume?)

L96: Typo: find instead of finding

L100: Recommend placing the sentence starting with "However" after eq 3 for clarity.

L114: Typo: should be "Kepler"

L125: Please specify what the 'beginning' of the year is defined as – a particular value of  $\lambda$ ?

L131: perhaps 'supports' instead of 'uses'

L159: maybe clarify that S\_0 can be manually set in the model as well

L241: clarify that it's +/- 1 Myr "of the present"

L322: "Even under hypothetical cases..." perhaps

**Figures**

**Figure 7 – please label axes on figure as well**

Author answer:

I made the recommended suggestions.

Figure 10 – I recommend putting all of the lines for each subplot onto the same axis and differentiating them perhaps by color. Right now, it's very difficult to see that the Be90 and Lask curves have a different eccentricity amplitude, due to the different y-axes used.

Author answer:

Thanks for warning me about the y-axis issues in Figure 10-a. I already fixed it by putting the same y-axis values. Another observation was to modify the dotted and

dashed line graphs with color graphs. In this case, an old-style line graph was adopted because I considered the readers with color view deficiencies. The GMD Journal recommends that the authors pay attention and be careful about it.

---

## Author Comment (AC3)

**Referee#1 - Feedback**

**Summary:**

**The author has built source code and a Graphical User Interface for calculating TOA incoming solar radiation based on orbital parameters. Other packages have been developed previously to do this, but the author's aim is for a more flexible (output more customized) and usable (for research and/or teaching) package. These calculations are useful for setting insolation boundary conditions in models, understanding the relative influences of individual orbital parameters on insolation, etc. A few major comments (below) relate to points of clarification in the description of DINSOL, comparison to other insolation calculators, and the code validation.**

1. **Flexibility: It would be helpful to provide more information comparing DINSOL with PALINSOL. I read the PALINSOL documentation and the biggest gains in flexibility in DINSOL seem to be in providing calculations for a 365-day calendar, as opposed to just a 360-day calendar, and at several time scales shorter than daily mean (e.g., 6 hour, 3 hour, 1 hour, 30 minutes or 15 minutes). Both programs allow adjustments to solar constant as well as access to various solutions for orbital parameters (Berger78, Berger90, Laskar04). One way in which PALINSOL seems to be more flexible is with specification of latitude. Any latitude value may be used, whereas in DINSOL, the user specifies the number of latitude bands and latitudes are assumed to be equally-spaced. Since some climate models use spectral grids (e.g., T42, T31) that are not equally-spaced, PALINSOL is more flexible in this regard. Also, PALINSOL could calculate insolation at the specific latitude of a paleoclimate proxy, another potential use case. At any rate, it would be helpful to the reader to provide a more detailed comparison such as this, so that they may choose the most appropriate tool for their application.**

Author answer:

The most significant DINSOL advantage compare to PALINSOL is its flexibility in preparing and writing data more easily. Please note that the DINSOL and PALINSOL are not competitors; both options have upsides and downsides and were developed for different purposes. The reason for developing the DINSOL started about three years ago while I was using the PALINSOL to prepare the boundary conditions for a simplified climate model. At this moment, I realized that it could be more complex than I expected because of the PALINSOL outputs format. Thus, I saw that the community needed a specific tool for preparing solar radiation for simplified and intermediate-complexity climate models. A more intuitive and flexible tool that should work as an ordinary program and still could have its source code adapted/modified easily by the users.

By the way, the PALINSOL is an awesome package; from it, I could improve a little bit more my comprehension of Berger and Milankovitch's works. Anyway, the primordial PALINSOL goal is to compute the Earth's orbitals parameters and the incoming solar radiation on the TOA, which requires the combination of some PALINSOL functions, where it is supposed that all users know how to manipulate these data using the R language. For instance, to compute the daily insolation adopting the PALINSOL package, we need to use the Insol function, which computes the mean daily insolation from a given true solar longitude and latitude. However, before that, we have to convert a 360-day calendar to his correspondent true solar longitude, which can be made using the function "day2l". Thus, In order to explain a little bit better, I wrote an R script to compute the daily insolation using the PALINSOL package:

```r
library("palinsol")

data1 <- ber78(0,degree=FALSE)
```

```r
for (day in 1:360){
  for (jlat in -90:90){
      tsl  <- day2l(data1,day)
      stoa <- data.frame(Insol(data1,long=tsl, lat=jlat*pi/180,S0=1365))
      print(paste(day,jlat,stoa[1,1,2]))
  }
}
```

As such, conform observed by Referee#1, calculating the mean daily insolation from PALINSOL is easy and just requires a small number of lines. It's important to mention that the users need to write a script for that, and we still have to use a function like data.frame to get the insolation data individually. On the other hand, DINSOL users don't need to spend time writing scripts or looking for the best R functionalities to record the results into a file. The DINSOL users can execute the program easily from just one command line or more intuitively from the graphical user interface. By the way, adopting an unequal latitude interval in the DINSOL is not an obstacle; the users must adapt the source code, which is easy.

**In general, what are the points where DINSOL should be considered more flexible than PALINSOL?**

1) Offering two different calendar options: 360 and 365 days.
2) Set as many lat-lon number points as the users intend to adopt.
3) Merging Laskar solutions (La04 and La10), which increase the time range from -249 Myr to 21 Myr.
4) Compute the instantaneous solar radiation globally at the TOA from five-time interval options.
5) All data results can be easily manipulated and accessed from external tools (R, Python, CDO, GrADS, NCLe, etc.).
6) Having some programming language skills is unnecessary to execute the DINSOL program standardly, where it can be executed from command lines or a graphical user interface.
7) The source code can be compiled and executed from different FORTRAN compilers, improving the DINSOL performance on the same machine.

Another upside of the DINSOL program is when the users intend to plot the annual daily insolation. Although the PALINSOL has a specific function to plot the daily insolation as a function of the true solar longitude, there are better approaches to visualize the annual insolation variability than the true solar longitude. The day of the year is preferable because we can see the orbital effect during the perihelion more clearly, as is expected by Kepler's laws of planetary motion. This is the reason of the DINSOL name. Below is the annual daily insolation considering a hypothetical Earth's eccentricity of 0.5. It's worth clarifying that the other orbital parameters are based on nowadays. Thus, the left figure was plotted using the DINSOL data (starting from January 1st), whereas the other was from the PALINSOL with Milankovitch function (starting from March 21st – Vernal Equinox).

[Figure]

The script used to plot the PAPLINSOL image is given below:

```r
library("palinsol")

**save figure [require png package]**
png("example_plot.png", width = 650, height = 550, type='cairo')

data1 <-ber78(0,degree=FALSE)

obl <- 23*(pi/180)
ect <- 0.5
prc <- 282*(pi/180)

orbit <- c(eps=obl, ecc=ect, varpi=prc)
M    <- Milankovitch(orbit)
plot(M, plot=contour, month=FALSE)
```

About the spectral grids, I agree with Referee#1 once the DINSOL was written to prepare the solar data using regular grids. Anyway, it's important to remember that it is common in a standard GCM to perform initially a pre-processing (Legendre transformation (LT), Fast Fourier Transformation (FFT), Spectral to grid conversion, or vice versa). Therefore, it's reasonable to imagine that high-complexity models already contain solar radiation routines, which makes the DINSOL and other similar tools unnecessary for most GCMs.

Speaking of which, the DINSOL was developed to support climate researchers working independently and using simplified or intermediate-complexity climate models. This approach is preferable during preliminary research analysis once the high-complexity models are harder to include new parameterizations without skilled team support. Therefore, the fact that DINSOL output data is given in a regular grid shouldn't be a downside. Moreover, it's worth mentioning that the users could convert the DINSOL output file into a Gaussian grid or still get the spectral coefficients by using some external tool or library.

2. **Usability: the code is well-documented and it was easy to run on a Unix system.  a) I was not able to test the GUI due to the number of dependencies it requires (fortran, python version 2.7, R, GrADS, as well as several additional libraries). The GUI looks like a useful tool for visualizing output, particularly in a classroom setting, and a future version that is easier to get running with fewer dependencies might be used more widely.  b) The two output formats are text and binary. Given the ubiquity of netcdf in climate modeling, adding this as an output option would be useful.  c) Another advantage of implementations such as PALINSOL is their modularity. For example, it is easy to write a loop in R using PALINSOL functions to calculate a transient time series of insolation, say through many thousands of years, for several specific latitudes, etc. Maybe the author could comment on how similar tasks would be completed with DINSOL. I imagine shell scripting would be part of the solution, but it doesn't seem as straightforward.**

Author answer:

Thanks for the suggestions and feedback. I was working to create a GUI executable file where the users no longer need to install as many library dependencies as previously. Now, users just need to install the WINE library, which is supported by many Linux distros and macOS. With regard to the data format, I tried to guarantee that the users did not get into trouble during the DINSOL source code compilation. A clean FORTRAN code without demanding external dependence is preferable to the success of the compilation process. Ironically, I should have had the same attention to the GUI dependencies, but it is better later than never.

Furthermore, the binary files generated by DINSOL are followed by descriptor files, which allow us to convert them to NetCDF by adopting some external tool. Despite that, the DINSOL already

brings some scripts to aid users in accessing the content of binary files by using the GrADS program or the R language. Regarding the NetCDF format conversion, I recommend adopting the Climate Data Operator (CDO). Your installation is supported on Linux machines and contains many functionalities.

For instance, if we intend to convert the binary files generated by each DINSOL simulation into NetCDF files, we only need to use the command line below:

**cdo -f nc import_binary my_data.ctl my_new_data.nc**

Below I brought a snapshot using this command on Ubuntu OS.

After getting the NetCDF file, the users could adopt many other CDO functions; where another useful example is the regular grid conversion into a Gaussian grid:

**cdo remapbil,t106grid radiation.nc t106_radiation_gaussian.nc**

Look at the result in the snapshot below:

Now, another interesting example is how to transform fields on a Gaussian grid to spectral coefficients, which reduce the file size dramatically:

**cdo gp2sp t106_radiation_gaussian.nc t106_radiation_spectral.nc**

Below I brought the last snapshot:

In summary, DINSOL users can easily convert the output data into NetCDF from an external tool. Furthermore, the DINSOL also provides a text file format because this file can be opened on any spreadsheet application (Excel, Calc, etc.), as well as some high-level programming languages (Python, R, etc.). Thus, imagining students at the beginning level, the text files are the best way to access the content in the output data directly.

Later, referee#1 also requested some opinion/explanation about the DINSOL to be used on long-term simulations, either orbital parameters or insolation. On this point, I agree that it is easier to compute the orbital parameters from the PALINSOL package, which demands a very simple and short script. Nevertheless, during the DINSOL evaluation, I had to modify the source code in order to get the orbital parameters from a time slice. In my humble opinion, it was also a simple task where I just needed to add a loop and print the orbital values in the console. Obviously, the other subroutines and console prints were inactivated, which was also easy. Thus, after compiling the modified source code, the user should save the console prints in a text file, for example:

**./dinsol.exe >orbital_time_slice.txt**

Below is given the same modifications adopted during the DINSOL evaluation.

```fortran
subroutine orbital_berger1978

!!!!!!!!!!!!!!!!!!!!!!!!!!!!!!
!                          !
!      BERGER 1978         !
!                          !
!!!!!!!!!!!!!!!!!!!!!!!!!!!!!!

use variable_names

integer :: t

print *,"     int     year        ecce_be78        oblq_be78        prcs_be78        esin_be78"

do t=1, 151

  year=1000*t-151000

  !Converting data from tables
  B78_A     =  B78_Amp1/3600.
  B78_ff    =  B78_Rate1*arcsec2rad
  B78_Phi   =  B78_Phase1*deg2rad
  B78_M     =  B78_Amp4
  B78_g     =  B78_Rate4*arcsec2rad
  B78_Beta  =  B78_Phase4*deg2rad
  B78_F     =  B78_Amp5*arcsec2rad
  B78_fp    =  B78_Rate5*arcsec2rad
  B78_Delta =  B78_Phase5*deg2rad

  !Spectral analysis
  eps       =  e_star78 + sum(B78_A*cos(B78_ff*year+B78_Phi))         !Obliquity
  esinpi    =  sum(B78_M*sin(B78_g*year+B78_Beta))                    !Eccentricity
  ecospi    =  sum(B78_M*cos(B78_g*year+B78_Beta))                    !Eccentricity
  psi       =  psibar78*year + zeta78 + sum(B78_F*sin(B78_fp*year+B78_Delta))   !General Precession
```

```fortran
    if (ecospi <= 0) then
        atanpi    = atan(esinpi/ecospi) + pi
    else
        atanpi    = atan(esinpi/ecospi)
    end if

    !Orbital parameters
    eps       = eps*deg2rad
    ecc       = sqrt(esinpi**2 + ecospi**2)
    varpi     = amod(atanpi+psi+pi,twopi)

    if (varpi < 0) varpi=varpi+twopi
        omega     = amod(varpi+pi,twopi)

    !Converting to degree
    oblq      = eps*rad2deg
    prcs      = varpi*rad2deg

    print *, t, year, ecc, oblq, prcs, ecc*sin(varpi)

end do

end subroutine orbital_berger1978
```

If the users have some skills with Fortran language, it would be a simple deployment. Otherwise, PALINSOL could be considered an alternative.

Now, computing solar radiation on long-term simulations can be a little more complex than the case of the orbital parameters because it involves the main program and subroutines to record the data. A shell script could be an alternative to avoid adapting the source code, and it's important to mention that there is already a shell script to perform multiple simulations in the folder output/.0K; this file is called replace.sh. This shell script could be easily adapted, and from an external tool like CDO would be possible to merge all data results in only one.

Despite the shell script solution, I already modified the DINSOL source code to work on long-term simulations. This modified version adopts parallelism from the OpenMP API, where the users can choose the year increment freely between the initial and final year. This version works with a simplified climate model (GREB), and I intend to prepare a new manuscript using long-term climate simulations by adopting both. When this new manuscript is concluded, I will submit it, and this modified DINSOL version will become public for anyone to download and execute.

Please note that any user can access the source code and the DINSOL descriptive preprint/manuscript, which must be enough to create derivate program versions.

3. **Description of DINSOL as a model: Strictly speaking, I think it is more accurate to refer to DINSOL as a "program" or "calculator" rather than a "model," and that DINSOL "computes" or "calculates" insolation rather than it "simulates" insolation. While there are some uncertainties in exact values of orbital parameters through time (and thus the Berger and Laskar solutions), once orbital parameter is specified, the equations translating orbital parameters to insolation are established and this becomes a computation or calculation rather than a simulation (where things have to be assumed and the answer is not exact). See for examples lines 225-227, but also many other places throughout the manuscript.**

Author answer:

Indeed, the DINSOL works as an ordinary program, where an ensemble of instructions is responsible for computing the results from a list of input parameters. Therefore, the word program is more appropriate; thanks for your feedback. Moreover, about the correct use of the words simulate/calculate, on this point, I'm afraid that calculating sounds accurate in the manuscript context, which is untrue, and I will explain why. Please, look at my following explanation; maybe it could clarify some doubts.

For instance, if we consider the solar radiation modeling area and all your branches, some assumptions are expected to make some specific parameterizations viable. Speaking of which, in the DINSOL program, the solar radiation is calculated/simulated, assuming that the vernal equinoxes always happen on March 21st, precisely starting from 00UTC. However, it's just a PMIP simplification, where we could also consider the autumnal equinox (September 23rd) as a fixed season date. Another example is that the daily insolation loop can assume precisely 365-day or 360-day calendars, whereas the tropical year is 365.2422 days.

Thus, in my view, unless the DINSOL adopted something like the **\*Borkowski (1996)** methodology, where the season dates are defined accurately, we shouldn't consider the DINSOL products as a precise computation of the real world. Therefore, the word simulation would correctly represent the DINSOL products in the climate modelling context, which does not mean an impediment to adopt the calculation/computation words. Moreover, the DINSOL author assumes no responsibility for data misuse.

*Borkowski, K. M.: The Persian Calendar for 3000 Years, Earth, Moon and Planets, 74, 223–230, https://doi.org/https://doi.org/10.1007/BF00055188, 1996.*

4. **Points of clarification related to DINSOL uses:   a) Title: "…tool to be coupled with climate models…" The word "coupled" within the context of modeling suggests a two-way flow of information. DINSOL would provide information to simplified climate models, but models would not give information back to DINSOL. I recommend changing this wording to something like: "tool for specifying solar radiation boundary conditions and for classroom use."   b) Line 60: "versatile tool ideal for paleoclimate simulations, such as those prepared on the PMIP" PMIP models already have code (internally) that calculate insolation from specified orbital parameters or from specified year, for the model time step and spatial grid, so DINSOL is not needed in that context.   c) The author mentions in several places that the ability to specify hypothetical orbital parameters is ideal for exoplanets. It is important to note, however, that DINSOL allows specification of only a 360-day or 365-day year, which are Earth specific.**

Author answer:

a) Indeed, the word coupled must be removed from the title and the manuscript text. Once the solar radiation at the TOA is an external forcing, it cannot be affected by climate simulations. Thus, I accept and thanks you for the feedback.

b) At the 60th line and in other parts of the manuscript, I tried to say that the users should be able to reproduce the PMIP simulations by adopting the DINSOL to prepare the insolation boundary conditions for simplified climate models. I will analyse how I can improve the text sentences to avoid misunderstands.

c) When I developed the source code, I made some tests using hypothetical orbital parameters and annual calendars. Please note that although the current version provides only two calendar options for an ordinary execution, users could easily include a new calendar option. Here, I will show the minimum steps to modify the source code and simulate the solar radiation from a new calendar option; in this example, I will adopt a year with 687 days, the Mars annual number of days. Please note that the functions to compute the seasons, perihelion, and aphelion dates need to be adapted to provide correct values. Despite that, daily insolation and instantaneous solar radiation run smoothly. I took some snapshots showing the changes in the source code. Here, I include a third option where the user will need to set it in the namelist file before executing the dinsol executable file.

Firstly, we need to modify the **dinsol.f90** file on the subroutine *dinsol_model.*

<table>
<tr><th>Standard code</th><th>Modified code</th></tr>
</table>

Second, we need to edit the **main.f90** file by adding a new calendar condition.

Lately, we need to set the calendar option number 3 in the namelist file, and after that, we just have to compile and execute the DINSOL program from the command lines.

Note that additional calendars weren't included because of the season dates functions, which demand detailed changes in the FORTRAN and GUI source codes. It should be a more suitable deployment in the subsequent program versions. By the way, once the PALINSOL users write scripts to simulate solar radiation, it would be easy for DINSOL users with some programming language skills.

Over the next page, I brought a plot of the Mars daily insolation. The setup adopted for this simulation is based on **\*Mendoza et al (2021)**, except for the calendar, where the authors adopted the true solar longitude instead of the number of days.

*Mendoza, Víctor M., Mendoza, Blanca, Garduño, René, Cordero, Guadalupe, Pazos, Marni, Cervantes, Sandro, & Cervantes, Karina. (2021). Thermodynamic simulation of the seasonal cycle of temperature, pressure and ice caps on Mars. Atmósfera, 34(1), 1-23. Epub 10 de febrero de 2021.https://doi.org/10.20937/atm.52747*

| Solar constant (W/m$^2$) | Calendar (days) | Eccentricity | Obliquity (dg) | Precession (dg) |
|---|---|---|---|---|
| 590 | 687 | 0.093 | 25.2 | 248 |

[Figure]

**Mars – DINSOL**

[Figure]

**Mars - Mendoza, Victor M. et al (2021)**

Therefore, as was looked above, from a small number of changes was possible to adapt the DINSOL program to simulate the Mars daily insolation. However, it's important to mention that instantaneous solar radiation requires a little more attention.

Firstly, we consider that the Earth's day contains 24 hours; thus, the time interval (6 hours, 3 hours, 1 hour, etc.) should be interpreted differently for other planets' setups. For instance, Jupiter's days during about 10 hours, then while a time interval of 6 hours is given from each 90° of Earth's longitude, the equivalent for Jupiter should be of 2.5 hours.

Secondly, the instantaneous solar radiation is simulated considering the direction of the planet's rotation. The Earth's rotation is given from the west to the east, which is represented by the signal plus in equation 21 conforms to the DINSOL preprint. This equation is presented here on the next page, and please note that an inverted direction rotation (westward) only demands substituting the plus signal with the minus signal.

$$h = -\pi + \left(\frac{2\pi}{nt}\right)(t_i - 1) + \left(\frac{2\pi}{nx}\right)(x_i - 1)$$

Finally, the –π value on the right side of the equation represents the initial time for each day, where -180° is the same that 00UTC. It's worth remembering that the descriptor files will work after the modifications. However, the time definition would still be based on Earth's calendar and time clock.

5. **Code validation:   a) There isn't a validation for orbital parameters calculated from the Be90 and Lask methods (Table 4) since GISS uses only Be78. It seems that a validation could be done against PALINSOL.   b) I'm not sure that the statistical tests in Tables 4 and 6 are a useful way to compare DINSOL with other calculators. The orbital parameter and insolation calculations should be exactly the same across calculators (as Table 4 shows) within rounding error. A U-test for whether the calculated medians are the same is not useful because it is not testing replication. I found RMSE to be most useful, and recommend the U-test results be deleted.   c) Astronomical dates Table 5, section 3.2. For clarification, these were not "modeled" by PMIP – they were calculated based on Berger78. Perhaps use something like the following for the Table 5 caption: "This table contains the dates … calculated by DINSOL and by PMIPII, both using the method of Be78, for ..." And, then the first sentence of section 3.2: "Table 5 contains the dates … aphelion calculated by DINSOL and by PMIPII, both using the method of Be78." And the third sentence of section 3.2: "by PMIPII" rather than "using PMIPII" since the PMIP team used Be78.  d) Monthly insolation Figure 11, section 3.3: The LGM differences shown in panel (i) have a pattern at high latitudes during spring and fall seasons. The colors are not randomly distributed as you would expect if the two calculations are different only within rounding error. Without a scale bar, I can't tell how large the differences are, but it is curious why this systematic bias exists.**

**a) Orbital parameters:**

I initially considered adopting the PALINSOL during the DINSOL evaluation, but a dataset already evaluated and published in some journal could be more reliable in the referee's opinion. Thus, once referee#1 has been recommended during the public discussion, I will include the PALINSOL orbital parameters (Be90, La04) in the evaluation.

**b) U test**

I agree. The U test will be removed from the statistical analyses section, maintaining the other statistics.

**c) Correct the text**

Thanks for the feedback. I'm following your observations regarding clarifying and fixing some sentences in section 3.2.

**d) Monthly insolation biases**

I added a scale by using a divergent color palette (blue to red) and performed the monthly insolation plot again, I was looking for some error in the DINSOL source code, but I didn't found. Thus, I plot a new Figure showing the DINSOL minus PMIPII for 360-day and 365-day calendars. The biases behavior for LGM (21K) is weird and let me confuse, but the values are too small, and the other periods (0K and 6K) conform to our expectations. Therefore, in order to try to understand the reason for the biases for 21K, I decided to make a plot of the PALINSOL minus PMIPII, and DINSOL minus PALINSOL.

[Figure]

The figure below shows that the PALINSOL minus PMIPII also provides systematic biases, such as the DINSOL. However, comparing the DINSOL and PALINSOL, the differences are just within the round error. Hence, it's reasonable to consider that there is some bias precisely in the PMIPII insolation data (21K). Please, if the referee agrees, I would like to include the PALINSOL monthly insolation (21K) in the model evaluation section. The idea is to clarify the PMIPII 21K biases.

[Figure]

**Minor comments:**

1. **"PMIPII" is referenced many times in the abstract and throughout the paper (e.g., lines 125, 129, 150, etc.) It is unclear why "PMIPII" is referenced rather than "PMIP" more generally. PMIP3 and PMIP4 have also used specified orbital parameters. The focus on PMIPII seems out of date. One exception is the validation section (section 3), in which calculations performed specifically by the PMIPII team were compared to DINSOL output.**

Author answer:

   I should use a more general name, without focus on phase II, except by the section 3 where additionally I will include a PALINSOL dataset. Your suggestion will be followed in the manuscript.

2. **Line 42: "From Messori et al. (2019), most climate model simulations showed the intensification and geographical expansion of the monsoonal precipitation during the mid-Holocene…" My understanding is that this paper presents results from one model only and that there are still large model-data discrepancies for mid Holocene monsoonal precipitation. This sentence is also quite specific in the context of this paragraph. To make a more general point about the importance of PMIP model-data comparisons and about recent advances in this area, I recommend one sentence summarizing and referencing the recent paper by Brierly et al.** https://doi.org/10.5194/cp-16-1847-2020

Author answer:

   Thanks for the paper recommendation. Well, in order to cater to the suggestion, I replaced the current citation (Messori et al. (2019)) per Brierley et al. (2020) conform below:

**Line 42:** *For instance, second Brierley et al. (2020), PMIP simulations suggest that during the mid-Holocene, the most pronounced and robust monsoonal changes occurred over northern Africa and the Indian subcontinent, where the simulated rain rate was 10% greater than the pre-industrial era (1850 CE).*

3. **Line 55: "none was developed to prepare ISR data flexibly" and "prepare custom solar radiation data." I would argue that PALINSOL is flexible and customizable to some degree, and this statement is too strong.**

Author answer:

   It looks like an overstatement due to the need for more context and information. As such, a more detailed paragraph could be:

**Line 55:** *Even though we have pre-existing programs to calculate the ISR following the Milankovitch cycles theory, they were developed to cater to different purposes and users. For instance, the PALINSOL package requires users to learn or already know to program in R language. Thus, the Earth-Orbit v2.1 could be considered an alternative to get around the R language requirement because it can be executed using a friendly GUI. Even still, it is expected that the Earth-Orbit v2.1 users buy a MATLAB license. Furthermore, another critical question is that none of them was idealized to prepare the solar radiation data as a boundary condition for climate models from only one command line or clique, saving user's time. Therefore, it would be helpful to have another software option that works similarly to the pre-existent tools. This software should be free and does not require programming language skills to execute ordinary tasks, such as globally computing the annual daily insolation.*

4. **Line 76: "modern day" rather than "current days"**

Author answer: It was fixed.

5. **Line 129: "typical climate models use a 360-day calendar" is not true, most of the PMIP climate models use a 365-day calendar. Perhaps it is typical for intermediate-complexity climate models to use a 360-day calendar and this could be clarified to make a distinction between different sorts of models.**

Author answer:

Now the sentence is specified as typical for intermediate-complexity climate models.

**Line 128**: PMIPII experiments use a 365-day and 360-day annual calendar, while typical intermediate-complexity climate models use a 360-day calendar.

6. **Line 257 and elsewhere: When referring to the PALINSOL software package, provide a citation and/or URL?**

Author answer:

I didn't cite PALINSOL correctly because there isn't a paper or document describing the package and providing one doi. In an email message, Michel Crucifix reported that he intends to publish an article describing the PALINSOL functions in The R Journal. However, looking for some immediate alternative, I found the PALINSOL hosted in the Zenodo repository. Now, I could cite the Michel Crucifix tool correctly by providing a standard doi. It should stay this way:

*Crucifix, M.: Palinsol: insolation for palaeoclimate studies, R package version 0.93, 2016. Doi: https://doi.org/10.5281/zenodo.7198534*

If this reference can be considered reliable, I will include it in the manuscript line 257. Thanks for warning me.

7. **Regarding the discussion on colormaps for Figure 11 and for the GUI: I agree with the previous reviewer that a divergent colormap is preferable for difference maps (e.g., "Comparison to Current"). Keeping the original yellow-blue colormap is preferable for the Daily Insolation and the Day Length plots since these are not differences.**

Author answer:

Concerning the color palettes, I have already fixed them by adopting a divergent color in the DINSOL GUI, such as the snapshot below:

[Figure]

About Figure 11, I wonder if it would be preferable to split it into three parts, such as sequence below:

1) A visual comparison between DINSOL and PMIPII contour fields for [0K], [6K-0K], and [21K-0K]. Showing both calendars: 360-day and 360-day.

[Figure]

2) The difference fields between DINSOL and PMIPII for [0K], [6K-0K], and [21K-0K] and keep both calendars: 360-day and 360-day.

[Figure]

3) The difference between PALINSOL and PMIPII [21K] and the difference between DINSOL and PALINSOL [21K], both for a 360-day calendar.

[Figure]

In my opinion, the evaluation could become clearer for readers.

**8. Scale bar for Figure 11 plots g-i is missing.**

Author answer:

Already explained previously.